# FLOW++: IMPROVING FLOW-BASED GENERATIVE MODELS WITH VARIATIONAL DEQUANTIZATION AND ARCHITECTURE DESIGN

## ABSTRACT

Flow-based generative models are powerful exact likelihood models with efficient sampling and inference. Despite their computational efficiency, flow-based models generally have much worse density modeling performance compared to state-of-the-art autoregressive models. In this paper, we investigate and improve upon three limiting design choices employed by flow-based models in prior work: the use of uniform noise for dequantization, the use of inexpressive affine flows, and the use of purely convolutional conditioning networks in coupling layers. Based on our findings, we propose Flow++, a new flow-based model that is now the state-of-the-art non-autoregressive model for unconditional density estimation on standard image benchmarks. Our work has begun to close the significant performance gap that has so far existed between autoregressive models and flow-based models.

## 1 INTRODUCTION

Deep generative models – latent variable models in the form of variational autoencoders (Kingma & Welling, 2013), implicit generative models in the form of GANs (Goodfellow et al., 2014), and exact likelihood models like PixelRNN/CNN (van den Oord et al., 2016b;c), Image Transformer (Parmar et al., 2018), PixelSNAIL (Chen et al., 2017), NICE, RealNVP, and Glow (Dinh et al., 2014; 2016; Kingma & Dhariwal, 2018) – have recently begun to successfully model high dimensional raw observations from complex real-world datasets, from natural images and videos, to audio signals and natural language (Karras et al., 2017; Kalchbrenner et al., 2016b; van den Oord et al., 2016a; Kalchbrenner et al., 2016a; Vaswani et al., 2017).

Autoregressive models, a certain subclass of exact likelihood models, achieve state-of-the-art density estimation performance on many challenging real-world datasets, but generally suffer from slow sampling time due to their autoregressive structure (van den Oord et al., 2016b; Salimans et al., 2017; Chen et al., 2017; Parmar et al., 2018). Inverse autoregressive models can sample quickly and potentially have strong modeling capacity, but they cannot be trained efficiently by maximum likelihood (Kingma et al., 2016). Non-autoregressive flow-based models (which we will refer to as "flow models"), such as NICE, RealNVP, and Glow, are efficient for sampling, but have so far lagged behind autoregressive models in density estimation benchmarks (Dinh et al., 2014; 2016; Kingma & Dhariwal, 2018).

In the hope of creating an ideal likelihood-based generative model that simultaneously has fast sampling, fast inference, and strong density estimation performance, we seek to close the density estimation performance gap between flow models and autoregressive models. In subsequent sections, we present our new flow model, Flow++, which is powered by an improved training procedure for continuous likelihood models and a number of architectural extensions of the coupling layer defined by Dinh et al. (2014; 2016).

## 2 FLOW MODELS

A flow model $f$ is constructed as an invertible transformation that maps observed data $\mathbf{x}$ to a standard Gaussian latent variable $\mathbf{z} = f(\mathbf{x})$, as in nonlinear independent component analysis (Bell &

Sejnowski, 1995; Hyvärinen et al., 2004; Hyvärinen & Pajunen, 1999). The key idea in the design of a flow model is to form $f$ by stacking individual simple invertible transformations (Dinh et al., 2014; 2016; Kingma & Dhariwal, 2018; Rezende & Mohamed, 2015; Kingma et al., 2016; Louizos & Welling, 2017). Explicitly, $f$ is constructed by composing a series of invertible flows as $f(\mathbf{x}) = f_1 \circ \cdots \circ f_L(\mathbf{x})$, with each $f_i$ having a tractable inverse and a tractable Jacobian determinant. This way, sampling is efficient, as it can be performed by computing $f^{-1}(\mathbf{z}) = f_L^{-1} \circ \cdots \circ f_1^{-1}(\mathbf{z})$ for $\mathbf{z} \sim \mathcal{N}(\mathbf{0}, \mathbf{I})$, and so is training by maximum likelihood, since the model density

$$\log p(\mathbf{x}) = \log \mathcal{N}(f_1 \circ \cdots f_L(\mathbf{x}); \mathbf{0}, \mathbf{I}) + \sum_{i=1}^{L} \log \left| \frac{\partial f_i}{\partial f_{i-1}} \right| \tag{1}$$

is easy to compute and differentiate with respect to the parameters of the flows $f_i$.

## 3 FLOW++

In this section, we describe three modeling inefficiencies in prior work on flow models: (1) uniform noise is a suboptimal dequantization choice that hurts both training loss and generalization; (2) commonly used affine coupling flows are not expressive enough; (3) convolutional layers in the conditioning networks of coupling layers are not powerful enough. Our proposed model, Flow++, consists of a set of improved design choices: (1) variational flow-based dequantization instead of uniform dequantization; (2) logistic mixture CDF coupling flows; (3) self-attention in the conditioning networks of coupling layers.

### 3.1 DEQUANTIZATION VIA VARIATIONAL INFERENCE

Many real-world datasets, such as CIFAR10 and ImageNet, are recordings of continuous signals quantized into discrete representations. Fitting a continuous density model to discrete data, however, will produce a degenerate solution that places all probability mass on discrete datapoints (Uria et al., 2013). A common solution to this problem is to first convert the discrete data distribution into a continuous distribution via a process called "dequantization," and then model the resulting continuous distribution using the continuous density model (Uria et al., 2013; Dinh et al., 2016; Salimans et al., 2017).

#### 3.1.1 UNIFORM DEQUANTIZATION

Dequantization is usually performed in prior work by adding uniform noise to the discrete data over the width of each discrete bin: if each of the $D$ components of the discrete data $\mathbf{x}$ takes on values in $\{0, 1, 2, \ldots, 255\}$, then the dequantized data is given by $\mathbf{y} = \mathbf{x} + \mathbf{u}$, where $\mathbf{u}$ is drawn uniformly from $[0, 1)^D$. Theis et al. (2015) note that training a continuous density model $p_{\text{model}}$ on uniformly dequantized data $\mathbf{y}$ can be interpreted as maximizing a lower bound on the log-likelihood for a certain discrete model $P_{\text{model}}$ on the original discrete data $\mathbf{x}$:

$$P_{\text{model}}(\mathbf{x}) := \int_{[0,1)^D} p_{\text{model}}(\mathbf{x} + \mathbf{u}) \, d\mathbf{u} \tag{2}$$

The argument of Theis et al. (2015) proceeds as follows. Letting $P_{\text{data}}$ denote the original distribution of discrete data and $p_{\text{data}}$ denote the distribution of uniformly dequantized data, Jensen's inequality implies that

$$\mathbb{E}_{\mathbf{y} \sim p_{\text{data}}} [\log p_{\text{model}}(\mathbf{y})] = \sum_{\mathbf{x}} P_{\text{data}}(\mathbf{x}) \int_{[0,1)^D} \log p_{\text{model}}(\mathbf{x} + \mathbf{u}) \, d\mathbf{u} \tag{3}$$

$$\leq \sum_{\mathbf{x}} P_{\text{data}}(\mathbf{x}) \log \int_{[0,1)^D} p_{\text{model}}(\mathbf{x} + \mathbf{u}) \, d\mathbf{u} \tag{4}$$

$$= \mathbb{E}_{\mathbf{x} \sim P_{\text{data}}} [\log P_{\text{model}}(\mathbf{x})] \tag{5}$$

Consequently, maximizing the log-likelihood of the continuous model on uniformly dequantized data cannot lead to the continuous model degenerately collapsing onto the discrete data, because its objective is bounded above by the log-likelihood of a discrete model.

### 3.1.2 VARIATIONAL DEQUANTIZATION

While uniform dequantization successfully prevents the continuous density model $p_{\text{model}}$ from collapsing to a degenerate mixture of point masses on discrete data, it asks $p_{\text{model}}$ to assign uniform density to unit hypercubes $\mathbf{x} + [0, 1)^D$ around the data $\mathbf{x}$. It is difficult and unnatural for smooth function approximators, such as neural network density models, to excel at such a task. To sidestep this issue, we now introduce a new dequantization technique based on variational inference.

Again, we are interested in modeling $D$-dimensional discrete data $\mathbf{x} \sim P_{\text{data}}$ using a continuous density model $p_{\text{model}}$, and we will do so by maximizing the log-likelihood of its associated discrete model $P_{\text{model}}(\mathbf{x}) := \int_{[0,1)^D} p_{\text{model}}(\mathbf{x} + \mathbf{u}) \, d\mathbf{u}$. Now, however, we introduce a dequantization noise distribution $q(\mathbf{u}|\mathbf{x})$, with support over $\mathbf{u} \in [0, 1)^D$. Treating $q$ as an approximate posterior, we have the following variational lower bound, which holds for all $q$:

$$\mathbb{E}_{\mathbf{x} \sim P_{\text{data}}} \left[ \log P_{\text{model}}(\mathbf{x}) \right] = \mathbb{E}_{\mathbf{x} \sim P_{\text{data}}} \left[ \log \int_{[0,1)^D} q(\mathbf{u}|\mathbf{x}) \frac{p_{\text{model}}(\mathbf{x} + \mathbf{u})}{q(\mathbf{u}|\mathbf{x})} \, d\mathbf{u} \right] \tag{6}$$

$$\geq \mathbb{E}_{\mathbf{x} \sim P_{\text{data}}} \left[ \int_{[0,1)^D} q(\mathbf{u}|\mathbf{x}) \log \frac{p_{\text{model}}(\mathbf{x} + \mathbf{u})}{q(\mathbf{u}|\mathbf{x})} \, d\mathbf{u} \right] \tag{7}$$

$$= \mathbb{E}_{\mathbf{x} \sim P_{\text{data}}} \mathbb{E}_{\mathbf{u} \sim q(\cdot|\mathbf{x})} \left[ \log \frac{p_{\text{model}}(\mathbf{x} + \mathbf{u})}{q(\mathbf{u}|\mathbf{x})} \right] \tag{8}$$

We will choose $q$ itself to be a conditional flow-based generative model of the form $\mathbf{u} = q_{\mathbf{x}}(\boldsymbol{\epsilon})$, where $\boldsymbol{\epsilon} \sim p(\boldsymbol{\epsilon}) = \mathcal{N}(\boldsymbol{\epsilon}; \mathbf{0}, \mathbf{I})$ is Gaussian noise. In this case, $q(\mathbf{u}|\mathbf{x}) = p(q_{\mathbf{x}}^{-1}(\mathbf{u})) \cdot \left| \partial q_{\mathbf{x}}^{-1} / \partial \mathbf{u} \right|$, and thus we obtain the objective

$$\mathbb{E}_{\mathbf{x} \sim P_{\text{data}}} \left[ \log P_{\text{model}}(\mathbf{x}) \right] \geq \mathbb{E}_{\mathbf{x} \sim P_{\text{data}}, \boldsymbol{\epsilon} \sim p} \left[ \log \frac{p_{\text{model}}(\mathbf{x} + q_{\mathbf{x}}(\boldsymbol{\epsilon}))}{p(\boldsymbol{\epsilon}) \left| \partial q_{\mathbf{x}} / \partial \boldsymbol{\epsilon} \right|^{-1}} \right] \tag{9}$$

which we maximize jointly over $p_{\text{model}}$ and $q$. When $p_{\text{model}}$ is also a flow model $\mathbf{x} = f^{-1}(\mathbf{z})$ (as it is throughout this paper), it is straightforward to calculate a stochastic gradient of this objective using the pathwise derivative estimator, as $f(\mathbf{x} + q_{\mathbf{x}}(\boldsymbol{\epsilon}))$ is differentiable with respect to the parameters of $f$ and $q$.

Notice that the lower bound for uniform dequantization – eqs. (3) to (5) – is a special case of our variational lower bound – eqs. (6) to (8), when the dequantization distribution $q$ is a uniform distribution that ignores dependence on $\mathbf{x}$. Because the gap between our objective (8) and the true expected log-likelihood $\mathbb{E}_{\mathbf{x} \sim P_{\text{data}}} \left[ \log P_{\text{model}}(\mathbf{x}) \right]$ is exactly $\mathbb{E}_{\mathbf{x} \sim P_{\text{data}}} \left[ D_{\text{KL}} \left( q(\mathbf{u}|\mathbf{x}) \,\|\, p_{\text{model}}(\mathbf{u}|\mathbf{x}) \right) \right]$, using a uniform $q$ forces $p_{\text{model}}$ to unnaturally place uniform density over each hypercube $\mathbf{x} + [0, 1)^D$ to compensate for any potential looseness in the variational bound introduced by the inexpressive $q$. Using an expressive flow-based $q$, on the other hand, allows $p_{\text{model}}$ to place density in each hypercube $\mathbf{x} + [0, 1)^D$ according to a much more flexible distribution $q(\mathbf{u}|\mathbf{x})$. This is a more natural task for $p_{\text{model}}$ to perform, improving both training and generalization loss.

### 3.2 IMPROVED COUPLING LAYERS

Recent progress in the design of flow models has involved carefully constructing flows to increase their expressiveness while preserving tractability of the inverse and Jacobian determinant computations. One example is the invertible $1 \times 1$ convolution flow, whose inverse and Jacobian determinant can be calculated and differentiated with standard automatic differentiation libraries (Kingma & Dhariwal, 2018). Another example, which we build upon in our work here, is the affine coupling layer (Dinh et al., 2016). It is a parameterized flow $\mathbf{y} = f_\theta(\mathbf{x})$ that first splits the components of $\mathbf{x}$ into two parts $\mathbf{x}_1, \mathbf{x}_2$, and then computes $\mathbf{y} = (\mathbf{y}_1, \mathbf{y}_2)$, given by

$$\mathbf{y}_1 = \mathbf{x}_1, \qquad \mathbf{y}_2 = \mathbf{x}_2 \cdot \exp(\mathbf{a}_\theta(\mathbf{x}_1)) + \mathbf{b}_\theta(\mathbf{x}_1) \tag{10}$$

Here, $\mathbf{a}_\theta$ and $\mathbf{b}_\theta$ are outputs of a neural network that acts on $\mathbf{x}_1$ in a complex, expressive manner, but the resulting behavior on $\mathbf{x}_2$ always remains an elementwise affine transformation – effectively, $\mathbf{a}_\theta$ and $\mathbf{b}_\theta$ together form a data-parameterized family of invertible affine transformations. This allows the affine coupling layer to express complex dependencies on the data while keeping inversion and

log-likelihood computation tractable. Using $\cdot$ and $\exp$ to respectively denote elementwise multiplication and exponentiation,

$$\mathbf{x}_1 = \mathbf{y}_1, \qquad \mathbf{x}_2 = (\mathbf{y}_2 - \mathbf{b}_\theta(\mathbf{y}_1)) \cdot \exp(-\mathbf{a}_\theta(\mathbf{y}_1)), \qquad \log\left|\frac{\partial \mathbf{y}}{\partial \mathbf{x}}\right| = \mathbf{1}^\top \mathbf{a}_\theta(\mathbf{x}_1) \qquad (11)$$

The splitting operation $\mathbf{x} \mapsto (\mathbf{x}_1, \mathbf{x}_2)$ and merging operation $(\mathbf{y}_1, \mathbf{y}_2) \mapsto \mathbf{y}$ are usually performed over channels or over space in a checkerboard-like pattern (Dinh et al., 2016).

### 3.2.1 EXPRESSIVE COUPLING TRANSFORMATIONS WITH CONTINUOUS MIXTURE CDFs

We found in our experiments that density modeling performance of these coupling layers could be improved by augmenting the data-parameterized elementwise affine transformations by more general nonlinear elementwise transformations. For a given scalar component $x$ of $\mathbf{x}_2$, we apply the cumulative distribution function (CDF) for a mixture of $K$ logistics – parameterized by mixture probabilities, means, and log scales $\boldsymbol{\pi}, \boldsymbol{\mu}, \mathbf{s}$ – followed by an inverse sigmoid and an affine transformation parameterized by $a$ and $b$:

$$x \longmapsto \sigma^{-1}\left(\text{MixLogCDF}(x; \boldsymbol{\pi}, \boldsymbol{\mu}, \mathbf{s})\right) \cdot \exp(a) + b \qquad (12)$$

$$\text{where} \quad \text{MixLogCDF}(x; \boldsymbol{\pi}, \boldsymbol{\mu}, \mathbf{s}) := \sum_{i=1}^{K} \pi_i \sigma\left((x - \mu_i) \cdot \exp(-s_i)\right) \qquad (13)$$

The transformation parameters $\boldsymbol{\pi}, \boldsymbol{\mu}, \mathbf{s}, a, b$ for each component of $\mathbf{x}_2$ are produced by a neural network acting on $\mathbf{x}_1$. This neural network must produce these transformation parameters for each component of $\mathbf{x}_2$, hence it produces vectors $\mathbf{a}_\theta(\mathbf{x}_1)$ and $\mathbf{b}_\theta(\mathbf{x}_1)$ and tensors $\boldsymbol{\pi}_\theta(\mathbf{x}_1), \boldsymbol{\mu}_\theta(\mathbf{x}_1), \mathbf{s}_\theta(\mathbf{x}_1)$ (with last axis dimension $K$). The coupling transformation is then given by:

$$\mathbf{y}_1 = \mathbf{x}_1, \qquad \mathbf{y}_2 = \sigma^{-1}\left(\text{MixLogCDF}(\mathbf{x}_2; \boldsymbol{\pi}_\theta(\mathbf{x}_1), \boldsymbol{\mu}_\theta(\mathbf{x}_1), \mathbf{s}_\theta(\mathbf{x}_1))\right) \cdot \exp(\mathbf{a}_\theta(\mathbf{x}_1)) + \mathbf{b}_\theta(\mathbf{x}_1)$$
$$(14)$$

where the formula for computing $\mathbf{y}_2$ operates elementwise.

The inverse sigmoid ensures that the inverse of this coupling transformation always exists: the range of the logistic mixture CDF is $(0, 1)$, so the domain of its inverse must stay within this interval. The CDF itself can be inverted efficiently with bisection, because it is a monotonically increasing function. Moreover, the Jacobian determinant of this transformation involves calculating the probability density function of the logistic mixtures, which poses no computational difficulty.

### 3.2.2 EXPRESSIVE CONDITIONING ARCHITECTURES WITH SELF-ATTENTION

In addition to improving the expressiveness of the elementwise transformations on $\mathbf{x}_2$, we found it crucial to improve the expressiveness of the conditioning on $\mathbf{x}_1$ – that is, the expressiveness of the neural network responsible for producing the elementwise transformation parameters $\boldsymbol{\pi}, \boldsymbol{\mu}, \mathbf{s}, \mathbf{a}, \mathbf{b}$. Our best results were obtained by stacking convolutions and multi-head self attention into a gated residual network (Mishra et al., 2018; Chen et al., 2017), in a manner resembling the Transformer (Vaswani et al., 2017) with pointwise feedforward layers replaced by $3 \times 3$ convolutional layers. Our architecture is defined as a stack of blocks. Each block consists of the following two layers connected in a residual fashion, with layer normalization (Ba et al., 2016) after each residual connection:

$$\text{Conv} = \text{Input} \rightarrow \text{Nonlinearity} \rightarrow \text{Conv}_{3 \times 3} \rightarrow \text{Nonlinearity} \rightarrow \text{Gate}$$
$$\text{Attn} = \text{Input} \rightarrow \text{Conv}_{1 \times 1} \rightarrow \text{MultiHeadSelfAttention} \rightarrow \text{Gate}$$

where Gate refers to a $1 \times 1$ convolution that doubles the number of channels, followed by a gated linear unit (Dauphin et al., 2016). The convolutional layer is identical to the one used by PixelCNN++ (Salimans et al., 2017), and the multi-head self attention mechanism we use is identical to the one in the Transformer (Vaswani et al., 2017). (We always use 4 heads in our experiments, since we found it to be effective early on in our experimentation process.)

With these blocks in hand, the network that outputs the elementwise transformation parameters is simply given by stacking blocks on top of each other, and finishing with a final convolution that increases the number of channels to the amount needed to specify the elementwise transformation parameters.

## 4 EXPERIMENTS

Here, we show that Flow++ achieves state-of-the-art density modeling performance among non-autoregressive models on CIFAR10 and 32x32 and 64x64 ImageNet. We also present ablation experiments that quantify the improvements proposed in section 3, and we present example generative samples from Flow++ and compare them against samples from autoregressive models.

Our experiments employed weight normalization and data-dependent initialization (Salimans & Kingma, 2016). We used the checkerboard-splitting, channel-splitting, and downsampling flows of Dinh et al. (2016); we also used before every coupling flow an invertible 1x1 convolution flows of Kingma & Dhariwal (2018), as well as a variant of their "actnorm" flow that normalizes all activations independently (instead of normalizing per channel). Our CIFAR10 model used 4 coupling layers with checkerboard splits at 32x32 resolution, 2 coupling layers with channel splits at 16x16 resolution, and 3 coupling layers with checkerboard splits at 16x16 resolution; each coupling layer used 10 convolution-attention blocks, all with 96 filters. More details on architectures, as well as details for the other experiments, will be given in a source code release.

### 4.1 DENSITY MODELING RESULTS

In table 1, we show that Flow++ achieves state-of-the-art density modeling results out of all non-autoregressive models, and it is competitive with autoregressive models: its performance is on par with the first generation of PixelCNN models (van den Oord et al., 2016b), and it outperforms Multiscale PixelCNN (Reed et al., 2017). As of submission, our models have not fully converged due to computational constraint and we expect further performance gain in future revision of this manuscript.

Table 1: Unconditional image modeling results

| Model family | Model | CIFAR10 bits/dim | ImageNet 32x32 bits/dim | ImageNet 64x64 bits/dim |
|---|---|---|---|---|
| Non-autoregressive | RealNVP (Dinh et al., 2016) | 3.49 | 4.28 | – |
| | Glow (Kingma & Dhariwal, 2018) | 3.35 | 4.09 | 3.81 |
| | IAF-VAE (Kingma et al., 2016) | 3.11 | – | – |
| | **Flow++ (ours)** | **3.09** | **3.86** | **3.69** |
| Autoregressive | Multiscale PixelCNN (Reed et al., 2017) | – | 3.95 | 3.70 |
| | PixelCNN (van den Oord et al., 2016b) | 3.14 | – | – |
| | PixelRNN (van den Oord et al., 2016b) | 3.00 | 3.86 | 3.63 |
| | Gated PixelCNN (van den Oord et al., 2016c) | 3.03 | 3.83 | 3.57 |
| | PixelCNN++ (Salimans et al., 2017) | 2.92 | – | – |
| | Image Transformer (Parmar et al., 2018) | 2.90 | 3.77 | – |
| | PixelSNAIL (Chen et al., 2017) | 2.85 | 3.80 | 3.52 |

### 4.2 ABLATIONS

We ran the following ablations of our model on unconditional CIFAR10 density estimation: variational dequantization vs. uniform dequantization; logistic mixture coupling vs. affine coupling; and stacked self-attention vs. convolutions only. As each ablation involves removing some component of the network, we increased the number of filters in all convolutional layers (and attention layers, if present) in order to match the total number of parameters with the full Flow++ model.

In fig. 1 and table 2, we compare the performance of these ablations relative to Flow++ at 400 epochs of training, which was not enough for these models to converge, but far enough to see their relative performance differences. Switching from our variational dequantization to the more standard uniform dequantization costs the most: approximately 0.127 bits/dim. The remaining two ablations both cost approximately 0.03 bits/dim: switching from our logistic mixture coupling layers to affine

coupling layers, and switching from our hybrid convolution-and-self-attention architecture to a pure convolutional residual architecture. Note that these performance differences are present despite all networks having approximately the same number of parameters: the improved performance of Flow++ comes from improved inductive biases, not simply from increased parameter count.

The most interesting result is probably the effect of the dequantization scheme on training and generalization loss. At 400 epochs of training, the full Flow++ model with variational dequantization has a train-test gap of approximately 0.02 bits/dim, but with uniform dequantization, the train-test gap is approximately 0.06 bits/dim. This confirms our claim in Section 3.1.2 that training with variational dequantization is a more natural task for the model than training with uniform dequantization.

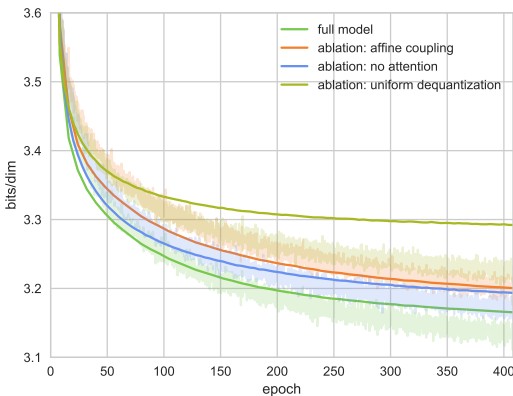

Figure 1: Ablation training (light) and validation (dark) curves on unconditional CIFAR10 density estimation. These runs are not fully converged, but the gap in performance is already visible.

Table 2: CIFAR10 ablation results after 400 epochs of training. Models not converged for the purposes of ablation study.

| Ablation | bits/dim | parameters |
|---|---|---|
| uniform dequantization | 3.292 | 32.3M |
| affine coupling | 3.200 | 32.0M |
| no self-attention | 3.193 | 31.4M |
| **Flow++ (not converged for ablation)** | **3.165** | **31.4M** |

### 4.3 SAMPLES

We present the samples from our trained density models of Flow++ on CIFAR10, 32x32 ImageNet, 64x64 ImageNet, and 5-bit CelebA in figs. 2 to 5. The Flow++ samples match the perceptual quality of PixelCNN samples, showing that Flow++ captures both local and global dependencies as well as PixelCNN and is capable of generating diverse samples on large datasets. Moreover, sampling is fast: our CIFAR10 model takes approximately 0.32 seconds to generate a batch of 8 samples in parallel on one NVIDIA 1080 Ti GPU, making it more than an order of magnitude faster than PixelCNN++ with sampling speed optimizations (Ramachandran et al., 2017). More samples are available in the appendix (section 7).

## 5 RELATED WORK

Likelihood-based models constitute a large family of deep generative models. One subclass of such methods, based on variational inference, allows for efficient approximate inference and sampling, but does not admit exact log likelihood computation (Kingma & Welling, 2013; Rezende et al., 2014; Kingma et al., 2016). Another subclass, which we called exact likelihood models in this work, does admit exact log likelihood computation. These exact likelihood models are typically

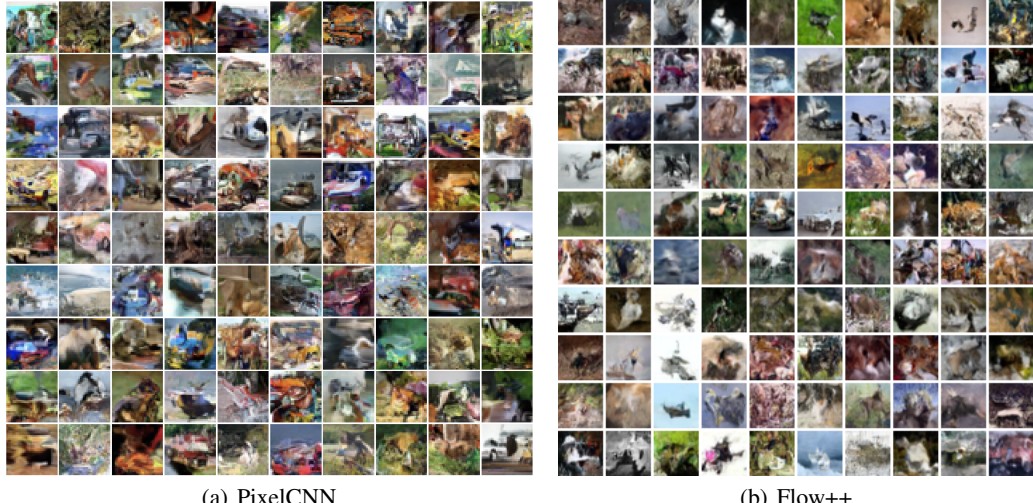

(a) PixelCNN                                    (b) Flow++

Figure 2: CIFAR 10 Samples. Left: samples from van den Oord et al. (2016b). Right: samples from Flow++, which captures local dependencies well and generates good samples at the quality level of PixelCNN, but with the advantage of efficient sampling.

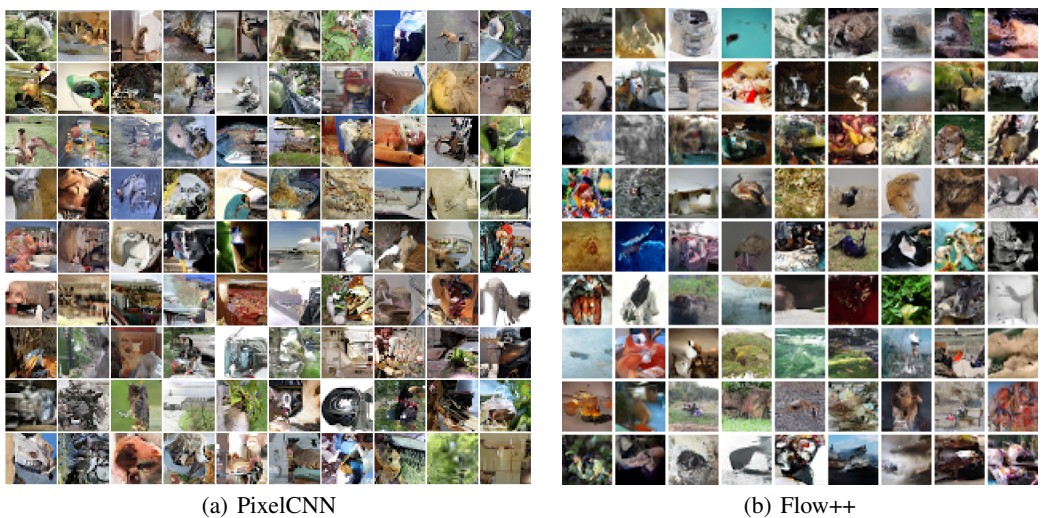

(a) PixelCNN                                    (b) Flow++

Figure 3: 32x32 ImageNet Samples. Left: samples from van den Oord et al. (2016b). Right: samples from Flow++. Note that diversity of samples from Flow++ matches the diversity of samples from an autoregressive model on this dataset, which is much larger than CIFAR10.

specified as invertible transformations that are parameterized by neural networks (Deco & Brauer, 1995; Larochelle & Murray, 2011; Uria et al., 2013; Dinh et al., 2014; Germain et al., 2015; van den Oord et al., 2016b; Salimans et al., 2017; Chen et al., 2017).

There is prior work that aims to improve the sampling speed of deep autoregressive models. The Multiscale PixelCNN (Reed et al., 2017) modifies the PixelCNN to be non-fully-expressive by introducing conditional independence assumptions among pixels in a way that permits sampling in a logarithmic number of steps, rather than linear. Such a change in the autoregressive structure allows for faster sampling but also makes some statistical patterns impossible to capture, and hence reduces the capacity of the model for density estimation. WaveRNN (Kalchbrenner et al., 2018)

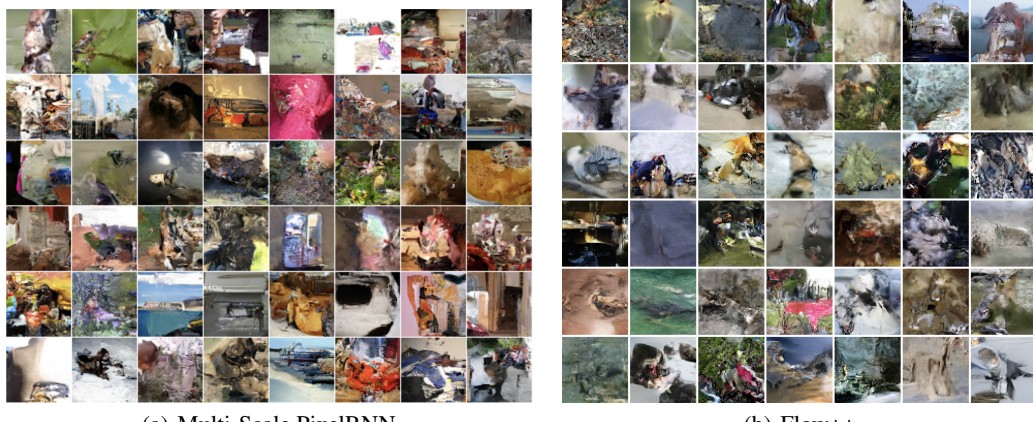

(a) Multi-Scale PixelRNN                                    (b) Flow++

Figure 4: 64x64 ImageNet Samples. Left: samples from Multi-Scale PixelRNN (van den Oord et al., 2016b). Right: samples from Flow++. The diversity of samples from Flow++ matches the diversity of samples from PixelRNN with multi-scale ordering.

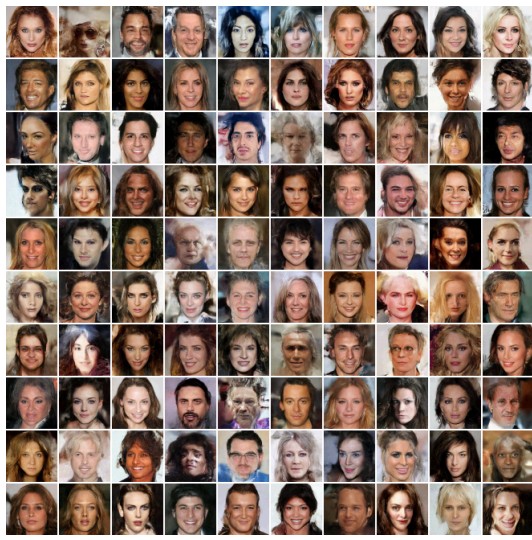

Figure 5: Samples from Flow++ trained on 5-bit 64x64 CelebA, without low-temperature sampling.

improves sampling speed for autoregressive models for audio via sparsity and other engineering considerations, some of which may apply to flow models as well.

There is also recent work that aims to improve the expressiveness of coupling layers in flow models. Kingma & Dhariwal (2018) demonstrate improved density estimation using an invertible 1x1 convolution flow, and demonstrate that very large flow models can be trained to produce photorealistic faces. Müller et al. (2018) introduce piecewise polynomial couplings that are similar in spirit to our mixture of logistics couplings. They found them to be more expressive than affine couplings, but reported little performance gains in density estimation. We leave a detailed comparison between our coupling layer and the piecewise polynomial CDFs for future work.

## 6 CONCLUSION

We presented Flow++, a new flow-based generative model that begins to close the performance gap between flow models and autoregressive models. Our work considers specific instantiations

of design principles for flow models – dequantization, flow design, and conditioning architecture design – and we hope these principles will help guide future research in flow models and likelihood-based models in general.

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

## 7 APPENDIX A: SAMPLES

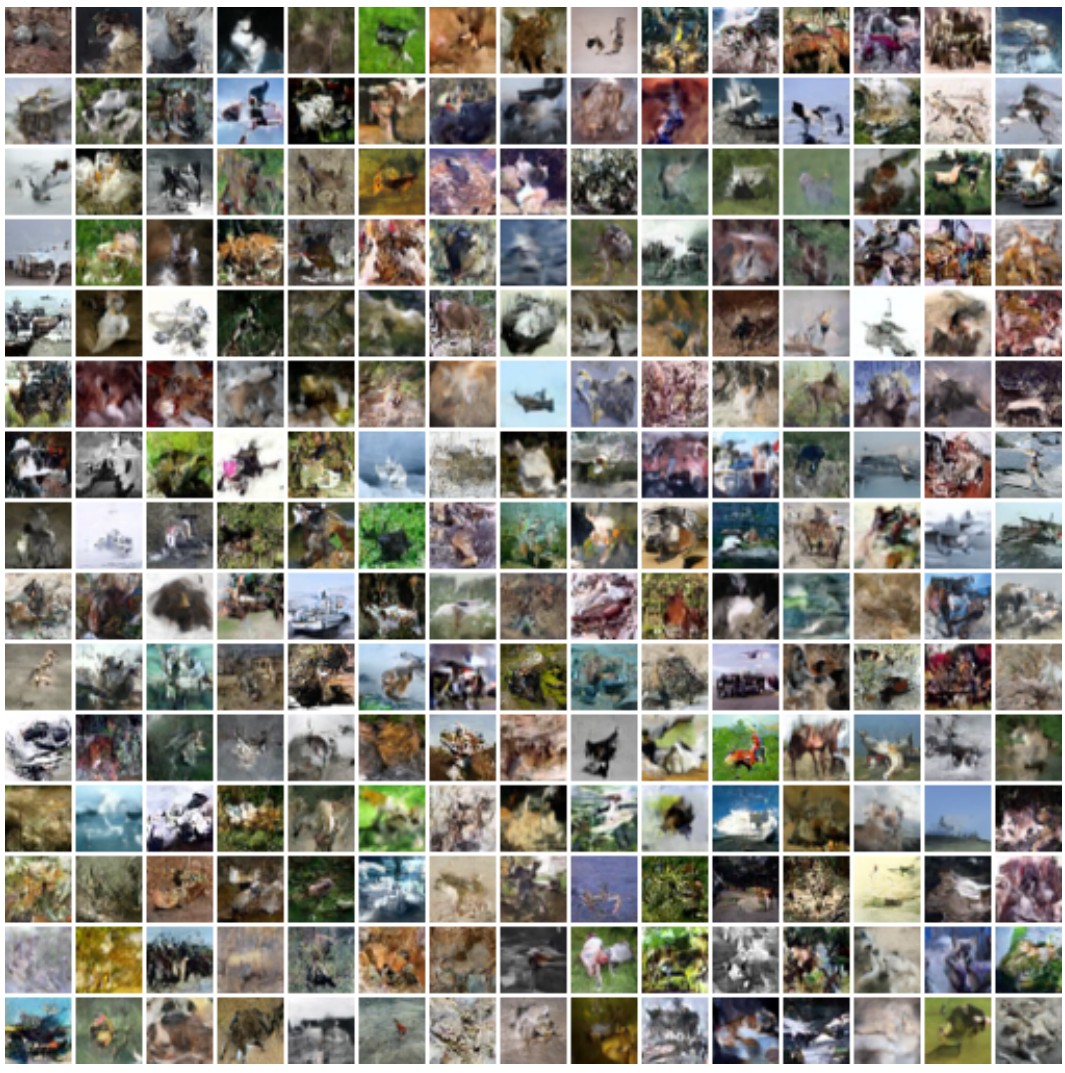

Figure 6: Samples from Flow++ trained on CIFAR10

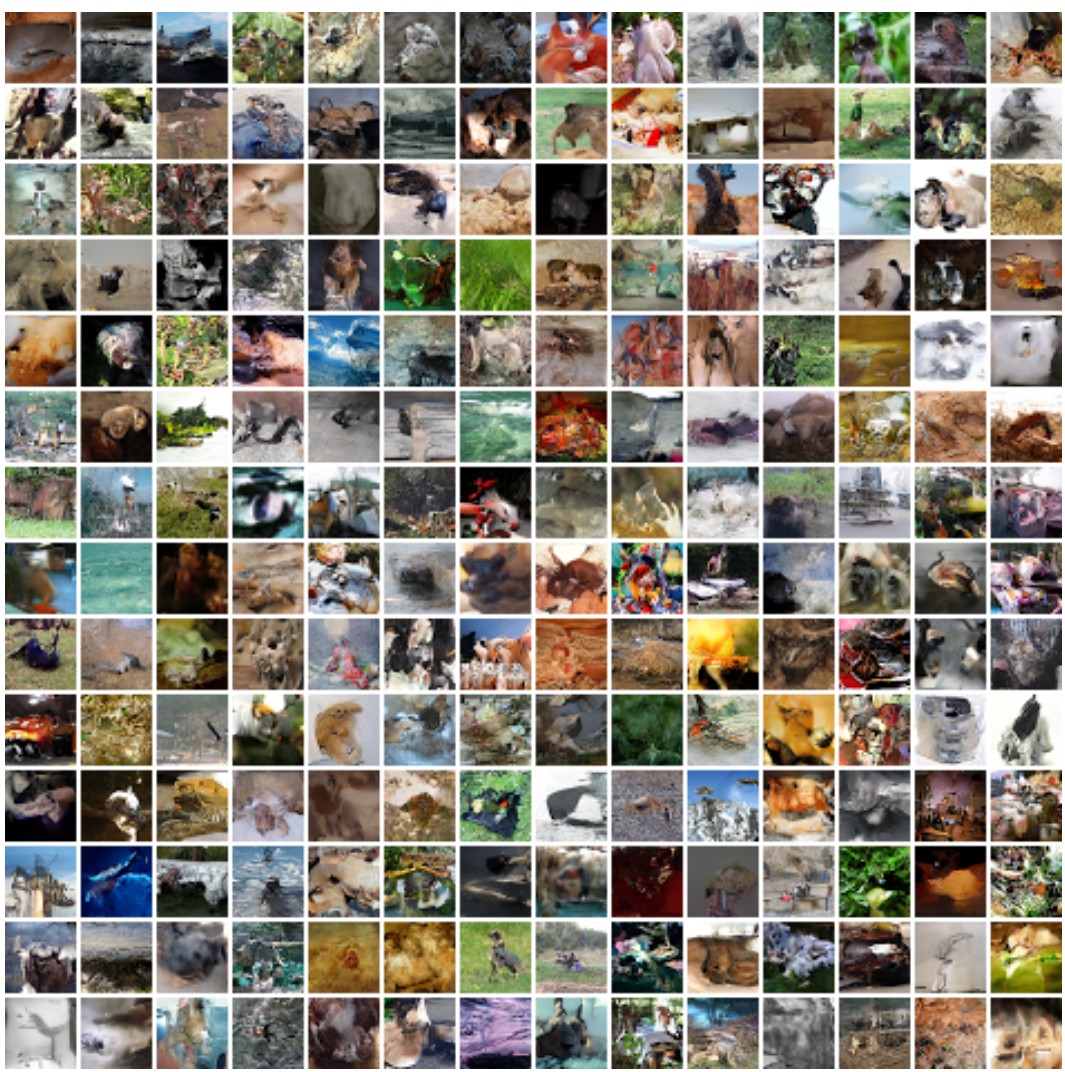

Figure 7: Samples from Flow++ trained on 32x32 ImageNet

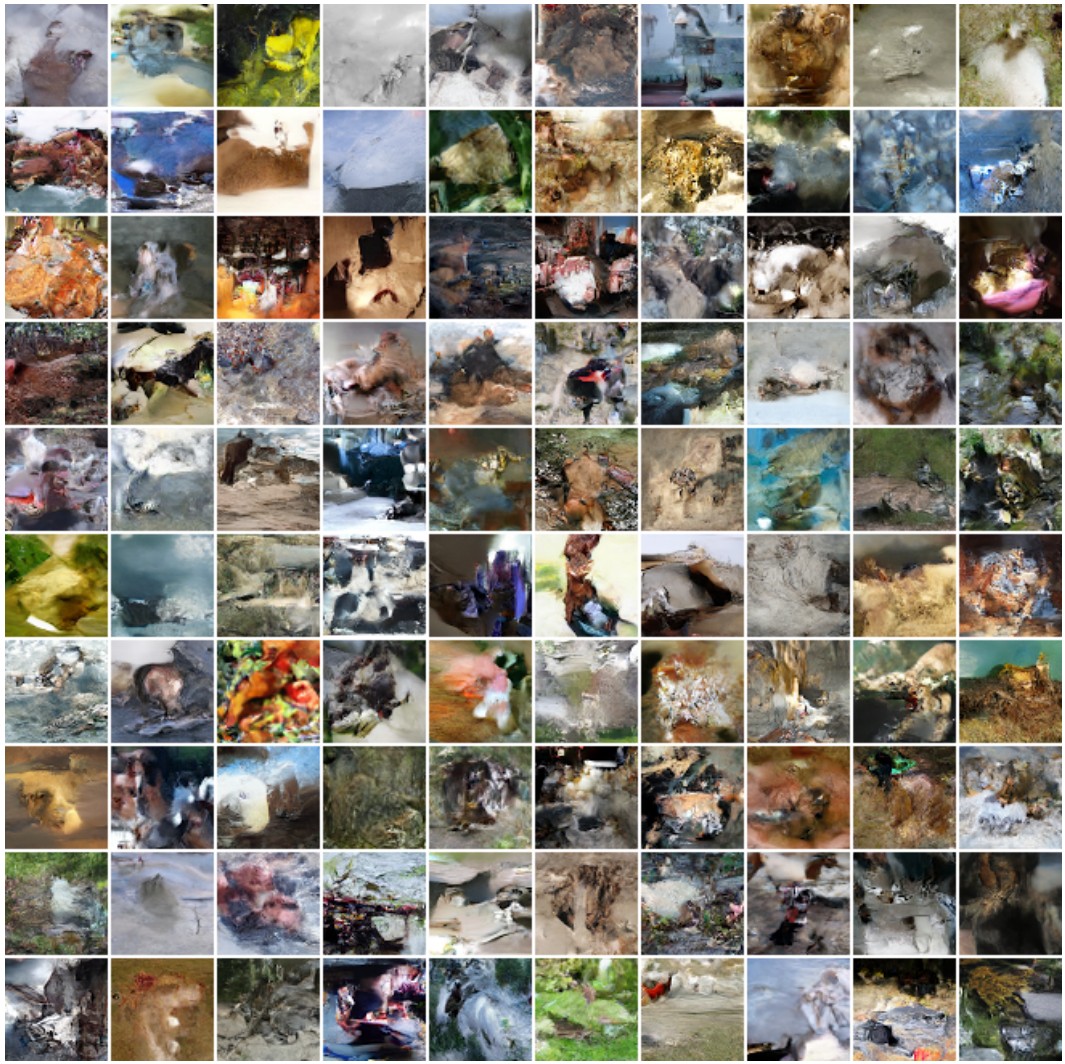

Figure 8: Samples from Flow++ trained on 64x64 ImageNet

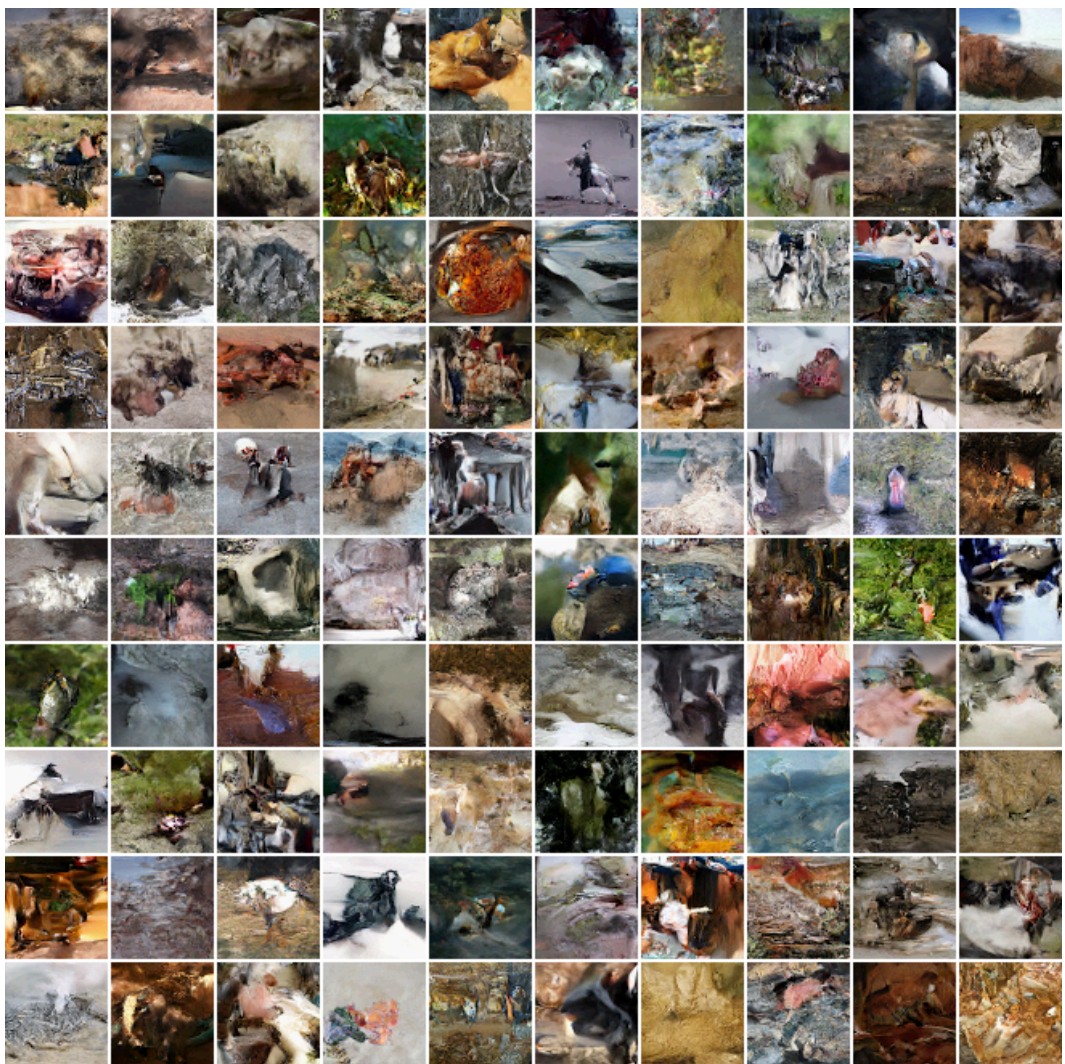

Figure 9: Samples from Flow++ trained on 5-bit 64x64 ImageNet

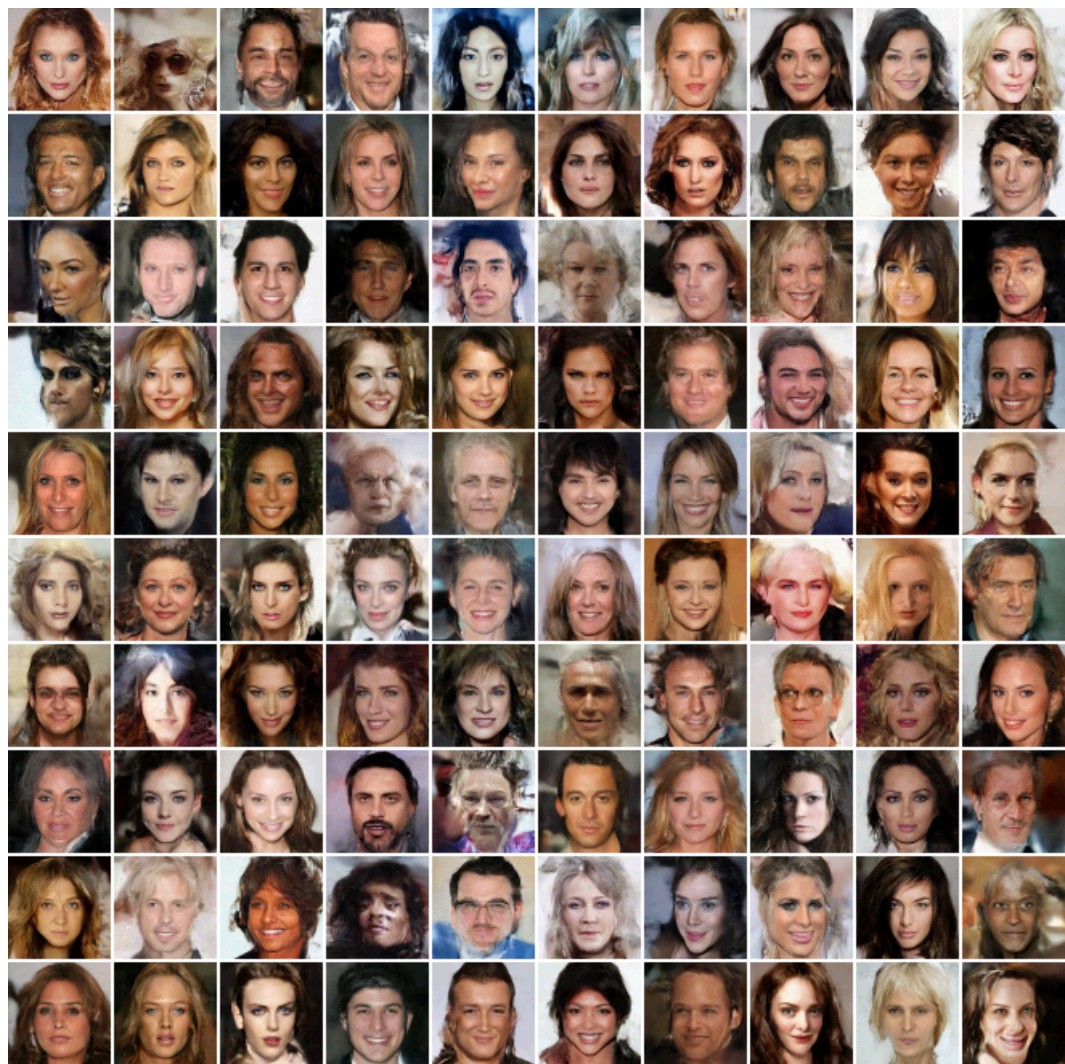

Figure 10: Samples from Flow++ trained on 5-bit 64x64 CelebA

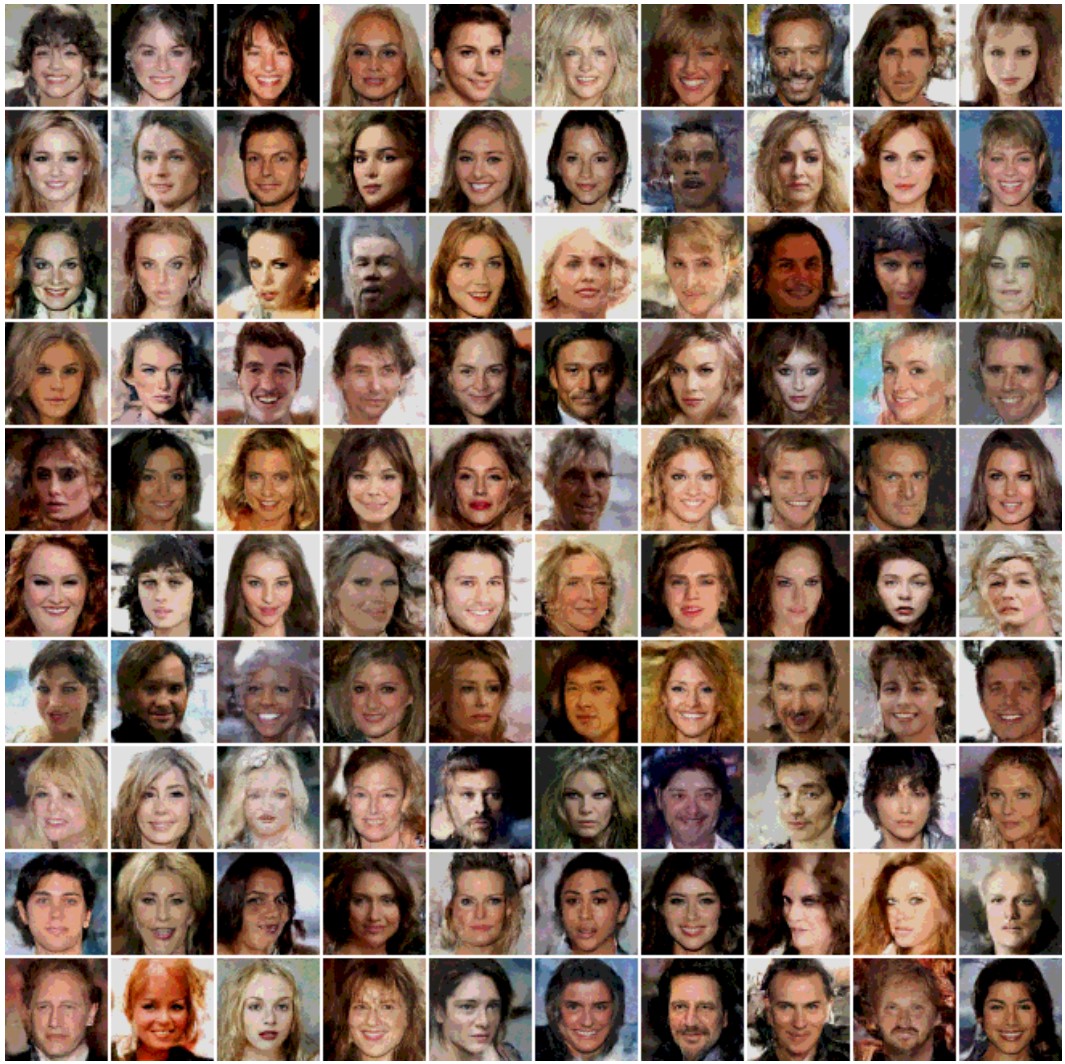

Figure 11: Samples from Flow++ trained on 3-bit 64x64 CelebA

