# OpenReview forum: "Flow++: Improving Flow-Based Generative Models  with  Variational Dequantization and Architecture Design  "
_ICLR.cc/2019/Conference_

### Official Review · AnonReviewer2 · 2018-10-24
**interesting improvements for RealNVP/Glow models, but not well analysed**

**Rating:** 5
**Confidence:** 5

**Review:**

The paper improves upon the Real NVP/Glow design by proposing better dequantization schemes and more expressive forms of coupling layers. I really like Real NVP models, which I think are a bit underappreciated. Thus, I’m happy that there are papers trying to improve their performance.  However, I wish this was done with more rigour.

The paper makes 3 claims about the current flow models: (1) it is suboptimal to use additive uniform noise when dequantizing images, (2) affine coupling layers are not expressive enough, and (3) the architectures fail to capture global image context. I’ll comment on these claims and proposed solutions below.

(1) I agree with the reasoning behind the need for a better dequantization distribution. However, I think the authors should provide an evidence that the lower bound is indeed loose when q is uniform. For example, for the CIFAR-10 model, the authors calculated a gap of 0.025 bpd when using variational dequantization. What would this gap be when using uniform q?  Maybe, a clear illustration of the dequantization effect on a simpler dataset or a toy example would be more useful.

(2) My main concern about the mixture CDFs coupling layer is how much bigger the model becomes and how much slower it trains. I find this analysis crucial when deciding whether 0.05 bpd improvement as reported in Table 1 is worth the hassle.

(3) As a person not familiar with the Transformer, I couldn’t understand how exactly self-attention works and how much it helps the model to capture the global image context. Also, I think this problem needs a separate illustration on a dataset of larger images.

The experiments section is very weak in backing up the identified problems and proposed solutions. Firstly, I think it is more clear if the ablation study is done in reverse: instead of making Flow++ and removing components, start with the vanilla model and then add stuff.  Secondly, it’s not clear if these improvements generalize across datasets, e.g. when images are larger than 32x32. Though, larger inputs may lead to huge models which are impossible to train when the resources are quite limited. That’s why I find it important to report how much complexity is added compared to the initial Real NVP. Also, I think it’s a well-known fact that sampling from PixelCNN models is slow unlike for Real NVPs, so I don’t find the results in Table 3 surprising or even useful.

To conclude, I find this paper unfinished and wouldn’t recommend its acceptance until the analysis of the problems and their solutions becomes better thought out.

---

> ### Author Response · Authors · 2018-11-26
> **Re: interesting improvements for RealNVP/Glow models, but not well analysed**
>
> - On looseness of the variational lower bound using a uniform q: our intention was not to emphasize the looseness of the bound, but rather to emphasize that the flow is forced to compensate for the inexpressive q by assigning high probability density to hypercubes around the data. This hurts generalization error, as it is an unnatural task for flows to perform: in our CIFAR10 ablation study, we found that the train-test performance gap with uniform q was 3 times larger than the gap using a flow-based q. We have updated the paper with discussion on this.
>
> - On training speed with the mixture-of-logistics layer: we found the difference in training speed to be negligible. When training our CIFAR model on 8 NVIDIA 1080ti GPUs with batch size 64, we achieved 110 images/sec without mixture of logistics, and 106 images/sec with mixture of logistics (with 32 mixture components).
>
> - We have updated our paper with illustrations and results on datasets with larger images: 64x64 ImageNet and 64x64 CelebA.
>
> - On importance of the individual model contributions: altogether, the model improvements we proposed make Flow++ the current state-of-the-art non-autoregressive model on CIFAR10, Imagenet 32x32, and ImageNet 64x64, and in fact it outperforms the Multiscale PixelCNN (Reed et al. 2017).

---

### Official Review · AnonReviewer3 · 2018-11-01
**Three ingredients for more powerful flow-based model**

**Rating:** 6
**Confidence:** 3

**Review:**

This paper offers architectural improvements for flow-based models that enable them to be very competitive with autoregressive models in terms of bits/dim metrics while still providing efficient sampling scheme. The three main contributions are the use of variational dequantization scheme, more powerful element-wise bijections (mixture of logistic CDF), and multi-head self-attention in the dependency structure.
The two first contributions are in my opinion the most interesting as:
- variational dequantization demonstrates the improvement that one can obtain by redefining part of the image processing that has been overlooked before;
- the inversion of element-wise bijection without closed form inverse can be efficiently approximated with bisection (binary search).
The performances achieved by the resulting model are in my opinion a stepping stone in the area of flow-based models and encouraging as to their potential.
The ablation study suggest that each contribution by themselves only improve slightly the model but that their simultaneous application results in a stronger boost in performance, which I can't explain from the paper. Nonetheless, some this ablation study was useful in tearing apart the contribution of each of several pieces of the model (missing pieces being gated convolutions, dropout, and instance normalization), although without explaining them.
Although flow-based model can intuitively sample faster than autoregressive models, the measure of sampling time is a bit interesting as an actual evidence of that claim. But the analysis of sampling time should be done on same hardware as to fair comparison before it can be a convincing argument.
Concerning variational dequantization, is there a reason coupling layer architecture was used instead of potentially more powerful model with less convenient inverses such as inverse autoregressive flow?

---

> ### Author Response · Authors · 2018-11-26
> **Re: Three ingredients for more powerful flow-based model**
>
> Regarding variational dequantization with IAF: the IAF is indeed a good candidate as a dequantization distribution, and is an interesting direction for future investigation.

---

### Official Review · AnonReviewer1 · 2018-11-02
**Three threads of improvements to normalizing flow models, reducing the gap between AR and non-AR models**

**Rating:** 6
**Confidence:** 4

**Review:**

I think the ideas are of sufficient interest to the community to merit acceptance & discussion, but I still miss the high resolution samples we got with the Glow paper. Responses to my concerns somewhat addressed, though simpler alternatives to uniform dequant would be nice.

=====

Improvements are attained on two image datasets by (a) variational dequantization, (b) mixture CDF coupling layers, and (c) self-attention in conditioning net.

Quality: The work is fine, demonstrating familiarity with recent work in flows and improving upon it. The experiments are on CIFAR-10 and 32x32 ImageNet. Unclear if the evaluation numbers are on a test set or a 'validation' set. I will be assuming test set. The visualizations are fine, but not nearly as convincing as the Glow visualizations on CelebA.

Clarity: The presentation is clear enough, and the motivation seems reasonable, though the assertion that all AR models are slow seems a bit belied by the recent WaveRNN work, which gets a Wavenet like model running in realtime on a phone. On the other hand, I felt like the proposed fixes were all a bit scattered here & there. Each could stand as a research topic on its own, and one paper can't fit in much analysis of all three. For example, a RealNVP style model usually needs to shuffle or reverse the channels to attain decent performance, but there's no discussion of how/whether that is done here. Folks wanting to replicate this work would want a formula for the tractable log-abs-det-jacobian of the coupling layer, but all we have is "involves calculating the pdf of the logistic mixtures".

Originality: Self-attention is not new, though its uptake in the conditioning networks of flow models has been slow/nonexistent. I found the dequantization improvement more novel. The new proposal for a coupling layer seems like a clever way of introducing more parameters in a structured manner.

Significance: Bringing flow models closer to the performance of AR models is good progress.


Questions
I wonder whether some kind of spline or cubic interpolation might achieve similar improvement over the uniform dequantization. Perhaps uniform is not the best baseline?
The new coupling layer might just be viewed as a way of introducing many more parameters in a structured manner. Have you compared parameter counts?
Appendix B shows some portion of the code, but seems like a missed opportunity to fit this into a framework like tfp.bijectors. The code seems glued in somewhat slapdash. For example, the tf_go function looks like debugging/logging code (unwanted), and lacks any usage.

I think this work is promising and interesting to the probabilistic modeling community, but needs some cleanup and some more compelling presentation (non image data? Glow-style graphics?).

---

> ### Author Response · Authors · 2018-11-26
> **Re: Three threads of improvements to normalizing flow models, reducing the gap between AR and non-AR models**
>
> - Our reported results follow the standard in likelihood-based generative modeling: the CIFAR10 results are on the test set, and the ImageNet results are on the publicly available validation sets, available here: http://image-net.org/small/download.php
>
> - Regarding speed of AR models: WaveRNN is indeed excellent work that increases AR sampling speed, and we expect that some of their improvements (such as weight sparsity) will also improve flow models. We look forward to seeing how well WaveRNN-like models perform on image datasets, which were the focus of our work.
>
> - We have updated the paper with CelebA results.
>
> - On checkerboard and channel splitting: we have updated the paper to mention how we use them, and details will be given in a cleaned source code release.
>
> - On whether our architecture is simply a matter of introducing more parameters: our ablations did control for parameter count, by increasing number of filters to compensate for removed parts of the architecture. We found that our improvements in density estimation were not explained by increased parameter count (in fact, some of our worse ablations have slightly more parameters than the full Flow++ model), but rather from improved inductive biases, and indeed our results are now state-of-the-art among non-autoregressive models on CIFAR10, 32x32 ImageNet, and 64x64 ImageNet. We have updated the section on ablations with this information.
>
> - On “spline or cubic interpolation instead of uniform dequantization”: we use uniform dequantization as our baseline since it is the standard dequantization technique employed in all prior work on continuous density modeling (see section 3.1 of the paper); we are not aware of any references in prior literature to spline or cubic interpolation for data dequantization.

---

### Meta-Review · Area_Chair1 · 2018-12-12
**Lovely main idea**

**Confidence:** 2
**Recommendation:** Reject

**Metareview:**

Strengths:
--------------
This paper was clearly written, contained novel technical insights, and had SOTA results.  In particular, the explanation of the generalized dequantization trick was enlightening and I expect will be useful in this entire family of methods.  The paper also contained ablation experiments.

Weaknesses:
------------------
The paper went for a grab-bag approach, when it might have been better to focus on one contribution and explore it in more detail (e.g. show that the learned pdf is smoother when using variational quantization, or showing the different in ELBO when using uniform q as suggested by R2).

Also, the main text contains many references to experiments that hadn't converged at submission time, but the submission wasn't updated during the initial discussion period.  Why not?

Points of contention:
-----------------------------
Everyone agrees that the contributions are novel and useful.  The only question is whether the exposition is detailed enough to reproduce the new methods (the authors say they will provide code), and whether the experiments, which meet basic standards, of a high enough standard for publication, because there was little investigation into the causes of the difference in performance between models.

Consensus:
----------------
The consensus was that this paper was slightly below the bar.